# An Empirical Study of Financial Efficiency and Stability of Shrimp–Mangrove Farming Model in Nam Dinh Province, Red River Delta, Vietnam

**Ngo Thanh Mai [1], Tran Thi Lan Huong [2,\*], Tran Tho Dat [3] and Dinh Duc Truong [1]**

[1] Faculty of Environmental, Climate Change and Urban Studies, The National Economics University (NEU), Hanoi 10000, Vietnam; thanhmai@neu.edu.vn (N.T.M.); truongdd@neu.edu.vn (D.D.T.)

[2] Faculty of Economics, The National Economics University (NEU), Hanoi 10000, Vietnam

[3] French Vietnamese Centre for Education Management (CFVG), The National Economics University (NEU), Hanoi 10000, Vietnam; tranthodat@neu.edu.vn

\* Correspondence: lanhuong@neu.edu.vn

**Abstract:** Shrimp–mangrove farming is a favorable production model suitable for the Red River Delta, Vietnam. This study was carried out in Nam Dinh province to evaluate the effectiveness and stability of the shrimp–mangrove farming model in the area. A participatory approach was carried out through surveys, focus group discussions, and interviews with 415 farming households in the Giao Thien and Giao An districts, the buffer zone of Xuan Thuy National Park, in 2022. We then used a cost–benefit analysis model to evaluate the financial performance and stability of the shrimp farming model. SWOT analysis was also used to identify opportunities and threats to this model. The research results showed that the shrimp–mangrove model has the advantages of low investment costs, diversified income sources, low risk, and environmental sustainability. However, the limitations of the model are low financial efficiency and not high stability. The main difficulties of the model are poor quality breeds, diseases, limited farming techniques of farmers, limited infrastructure system, the impacts of climate change, and low productivity. The study also proposes management implications to enhance the effectiveness and sustainability of shrimp farming in the Red River Delta, Vietnam.

**Keywords:** cost benefit analysis; sustainability; SWOT analysis; productivity; effectiveness

## 1. Introduction

Shrimp is one of the seafood products with high economic value, earning foreign currency from exports and creating livelihoods for most farmers in the coastal areas of Vietnam [1,2]. Out of the country's total area and production of shrimp, the Red River Delta comprises 48.2% of the area and contributes 24.8% of Vietnam's farmed shrimp production [3]. In addition to intensive and semi-intensive farming, the popular form of shrimp farming is still performed extensively and has increased in combination with mangrove planting. Previously, this form of farming accounted for about 87% of the total shrimp farming area and produced 52% of total shrimp production [4,5]. Currently, climate change is increasing the frequency of extreme weather events such as storms, floods, heat waves, and droughts. Hence, mangrove ecosystems have become especially important in mitigating the adverse effects of climate change [6,7].

The Red River Delta is vulnerable to the impacts of climate change, including agricultural production, livelihoods, and mangrove ecosystems. Shrimp farming in the Red River Delta, in particular and across the country in general, is heavily affected by climate change; unseasonal rains and extreme heat with increasing frequency have caused significant damage to shrimp farmers [8]. The trend of integrated shrimp farming, such as shrimp–rice and shrimp–mangrove models, is considered an intelligent farming solution to adapt to climate change [9]. Typically, the area of shrimp–mangrove farming in the Red

River Delta region accounts for about 185,000 ha, of which the water surface area accounts for 37,000 ha [10]. The outstanding features of the shrimp–mangrove model are low risk, low investment, and safe for the environment. However, the model's stability and financial efficiency constantly fluctuate and change depending on the ecological sub-region, farming method, and capacity of farmers.

Stemming from the above context, this study was conducted to assess the efficiency and financial stability of the shrimp–mangrove model in the Red River Delta, thereby determining the factors affecting the model and proposing solutions to improve the incomes and livelihoods of local farmers.

## 2. Study Site

The study was conducted in the Giao Thien and Giao An districts located in the buffer zone of Xuan Thuy National Park (XTNP), Nam Dinh province, Northern Vietnam (Figure 1).

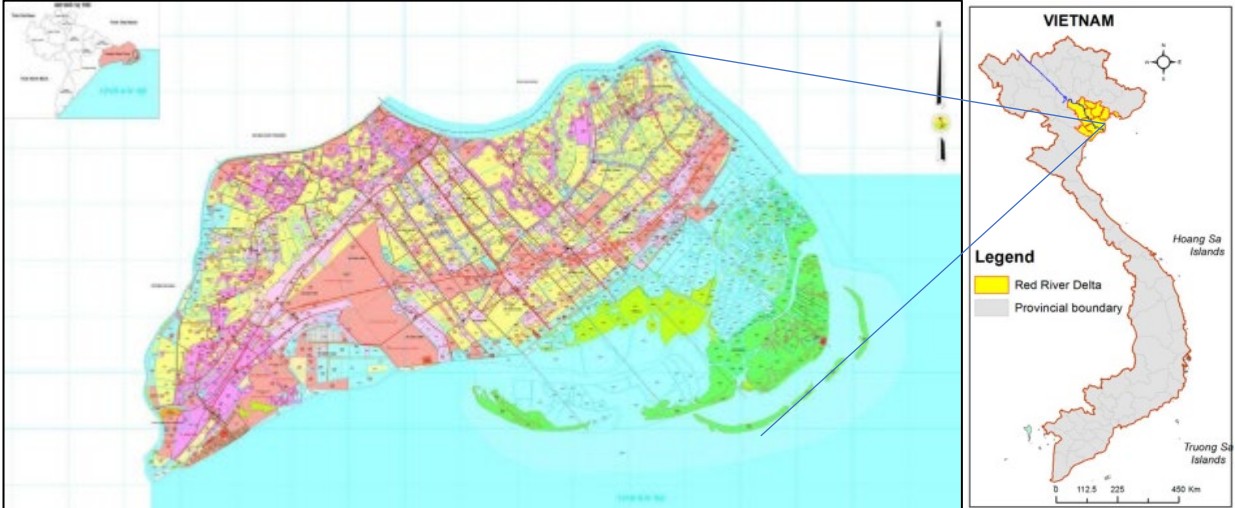

**Figure 1.** The study site in Nam Dinh province, Vietnam.

XTNP is a large alluvial area located south of the mouth of the Red River, about 150 km southeast of Hanoi. The fertile alluvium of the Red River and the sea has created a wetland with many rare species of wildlife and migratory birds [11].

XTNP has the southern boundary of the Vop River in the Giao Thuy district, Nam Dinh province. The park comprises a 7100 ha conservation area, including the Con Lu, Con Ngan, and Con Xanh alluvial areas, and a buffer zone of 8000 ha, including nearly 3000 ha of mangroves. Con Ngan is the largest in the National Park, with aquaculture lagoons and mostly covered with mangroves. Con Lu consists of a wide sandy beach, alluvial flats, and a small area of aquaculture ponds. Con Xanh is the smallest and still accretes alluvium from the Red River. Con Lu and Con Xanh are often flooded at high tide [7,10].

Within the area of the National Park, the highest point is only 3 m and the deepest water level is about 6 m. It is home to many valuable aquatic species, wild flora and fauna, and especially rare migratory birds. XTNP has special environmental conditions and is an intermediate ecosystem between aquatic and terrestrial ecosystems as well as freshwater and saltwater aquatic ecosystems. Therefore, this place has developed a population of both terrestrial and aquatic fauna and flora. There are many rare species of flora and fauna listed in various Red Books [3,12].

In January 1989, Xuan Thuy was the first wetland of Southeast Asia to join the RAMSAR International Convention. In January 2003, the Prime Minister signed Decision No. 01/2003/QD-TTg officially upgrading Xuan Thuy Wetland Nature Reserve to XTNP. In December 2004, UNESCO recognized XTNP as the core area of the World Biosphere Reserve in the inter-coastal region.

The context of the formation and development of the shrimp–mangrove model in districts is different due to the ecological, natural, and human conditions. However, there are some basic features in common. Accordingly, in the 1970s, the land where the shrimp–mangrove model formed was natural mangrove land (nipa palm) that had not been exploited. By the mid-late 1980s, the defense dyke system was built according to the national defense policy in coastal areas. This was also the timeline for forming two ecological sub-regions inside and outside the dyke [12].

When the policy of forest land allocation was implemented in 1995 by the government and further detailed in 2005, the shrimp–mangrove model evolved accordingly (Thuy et al., 2021). In five buffer zone districts, people have developed ecological shrimp farming models. In these models, farming ponds are dug with mangroves planted in the ponds, taking water and draining it naturally from the outside. There are two main types of shrimp–mangrove models: (1) the covering form (with the dyke surrounding the shrimp square, the mangrove planted in large numbers in the middle of the shrimp square, and the ditch/pond around the mangrove), (2) the riparian form (with mangroves planted on the ridge and interspersed in the ditch/shrimp pond, but a border still surrounds the mangroves) (Figure 2). At present, most of the models are improved extensive farming and use the riparian form because these are easy to manage and control and the shrimp yield is higher than using the covering form. Because the riparian form helps to distribute the shrimp density evenly, access to natural food sources is better than in the covering form [13].

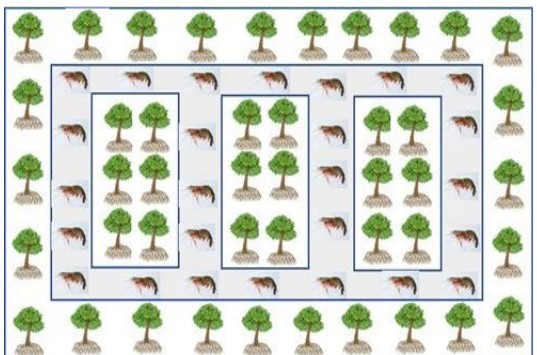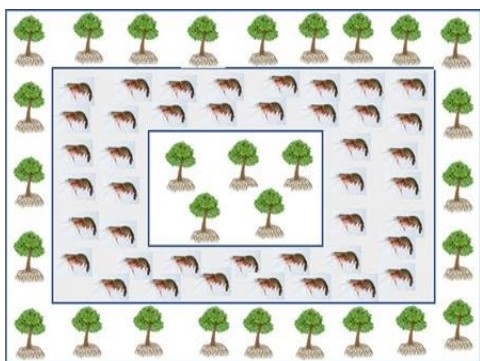

**Figure 2.** The riparian form (**left**) and the covering form (**right**). Source: Authors' compilation (2022).

### 3. Methodology and Data

*3.1. Financial Efficiency Model*

This study used cost–benefit analysis (CBA) to analyze the financial efficiency and stability of the shrimp–mangrove model in Nam Dinh province, Vietnam. CBA is a technique that helps businesses and households determine the benefits and costs of projects or activities so that they can make decisions about how to use scarce resources [14,15]. Several studies have been implemented to analyze the costs and benefits of farming systems [15–18]. In this research, we adapted the CBA methods of Charleson (2020) and Tengku (2021) for analysis [17,18]. Specifically, we used the following economic models to calculate the financial efficiency and stability of the shrimp–mangrove farming model in the study area.

(I) Total revenue represents the total income farming households per season and is determined by the formula:

$$TR = P \times Q$$

*TR*: total revenue (thousand VND/hectare/year)
*P*: market price of shrimp (thousand VND/kg/year)
*Q*: yield of shrimp (kg/hectare/year)

(II) Total cost of production is given by:

$$Total\ cost = Fixed\ cost + Variable\ cost$$

*Variable costs* are costs incurred along with production, including seed costs, labor, fertilizers, harvesting costs, pond preparation costs, and other costs.

*Fixed costs* are costs that do not depend on production output but create a premise for production, including land tax and depreciation of fixed assets.

(III) Gross profit of production is determined by:

$$GP = TR - TC$$

*GP*: gross profit (thousand VND/hectare/year)

(IV) Benefit–cost ratio (*BCR*) is the final indicator for comparing the economic efficiency of shrimp farming. BCR/hectare was calculated by dividing the total revenue for the total cost of production:

$$BCR_T = (TR/TC)_T$$

*BCR*: benefit/cost ratio per hectare

To evaluate the differences in cost, revenue, and profit between the two shrimp farming models, we used the independent samples t-test to test the differences in the mean values of these variables using SPSS software.

To study the factors affecting the financial performance of the shrimp–mangrove model, we used and adapted production theory and production function [19–21]. According to Suwannakit (2016), production is the process of combining various inputs, both material and immaterial, in order to create output. In the production process, the production function shows the relationship between the maximum amount of product that can be produced and the amount of factor input with a certain level of knowledge related to the technology. In short, the raw materials in production are classified as follows: capital, land, labor or natural resources, and at the same time, finished goods are transformed from raw materials through the production process [22,23]. Depending on the nature of the business, they can be variable or fixed. The production function creates the relationship between outputs and inputs. In the production process, the effectiveness of the relationship between output and input depends on the different quantities used, and they also depend on the yield at each point of output quantity. It is common practice that several forms of controllable inputs are used to achieve the output of a product. The production function assesses the relationship between the inputs and the quantity of output.

From the production function model, many empirical studies are adapted to evaluate the economic efficiency of production activities and the factors affecting efficiency [23–26]. Studies using multiple regression to evaluate the factors affecting financial performance are used quite commonly. Stemming from the research objective, after examining previous studies and the characteristics of the study area, the proposed explanatory variables to be included in the analysis of the correlation with the profitability of the shrimp–mangrove model in the study area are shown in Table 1 and Figure 3.

Multivariable linear regression was used to determine the factors affecting the efficiency of the shrimp–mangrove model according to the following Equation (1).

$$Y = \beta_0 + \beta_1 X_1 + \beta_1 X_1 + \beta_2 X_2 + \beta_3 X_3 + \ldots + \beta_n X_n \tag{1}$$

In which:

*Y* is the dependent variable or model profit (million VND/ha/year)
$\beta_0$ is a constant
$\beta_1$, $\beta_2$, $\beta_3$, $\ldots$, $\beta_n$ are regression coefficients that show the effect of each variable *X* on the value of variable *Y* when the other variables remain constant
$X_1$, $X_2$, $X_3$, $\ldots$, $X_n$ are the explanatory variables for the variable *Y*
$\varepsilon$ is the random error for which the value is assumed to follow a probability distribution

**Table 1.** The explanatory variables included in the analytical model.

| Explanatory Variables | Symbol | Description and Codes |
|---|---|---|
| Shrimp square manager obtained technical training | $X_1$ | 1 = being trained, 0 = not being trained |
| Price of shrimp breeds (size) | $X_2$ | Price at the time of stocking/raising (VND per shrimp) |
| Production contract | $X_3$ | 1 = yes, 0 = no (between farmers and enterprises in terms of input supply/output purchase) |
| Amount of shrimp breeds stocked | $X_4$ | Number of shrimp/ha/crop (year) |
| Number of shrimp stockings | $X_5$ | Number of stockings/crop (year) |
| Amount of crab breeds stocked | $X_6$ | Number of crabs/ha/crop (year) |
| Price of crab breeds (size) | $X_7$ | Price at the time of stocking/raising (VND per shrimp) |
| Shrimp square manager's production experience | $X_8$ | Number of years of direct management and cultivation of shrimp-mangrove model |

Source: Author's compilation.

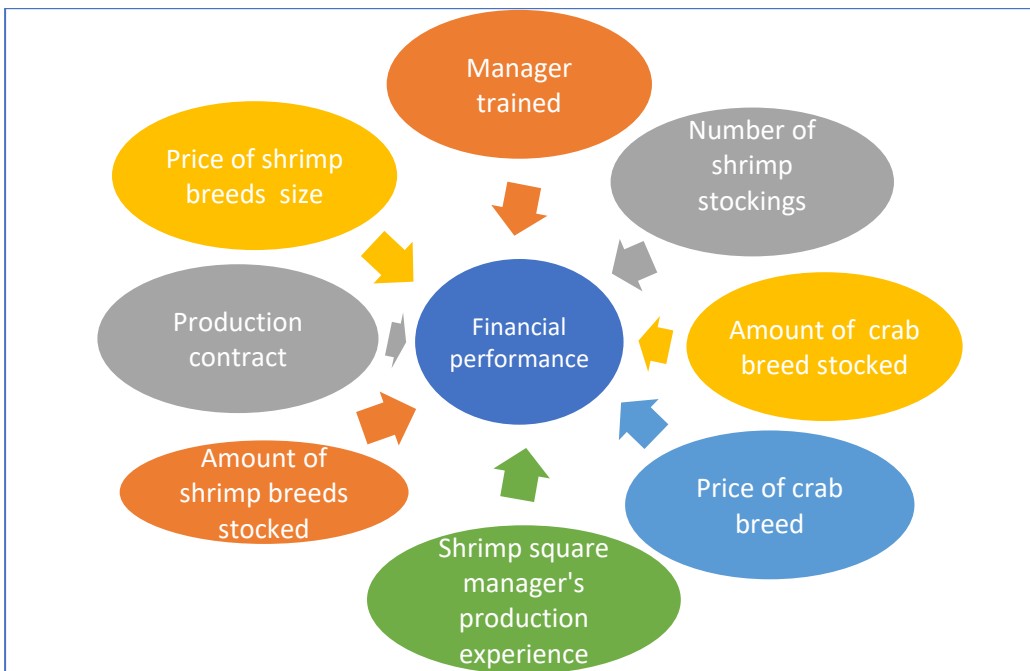

**Figure 3.** Model of factors affecting financial performance of shrimp farming.

*3.2. Evaluate the Financial Stability of the Model*

The stability of the model was evaluated based on the coefficient of variation (CV) according to the formula CV = SD/X, where:

CV: coefficient of variation
SD: standard deviation
X: average value

Stability refers to the volatility (change) more or less with external influences on the production system [27,28]. In actual production, the stability of the shrimp–mangrove model is often considered through the criteria investment cost, yield, and selling price because these criteria directly affect the profit and financial efficiency of the shrimp–mangrove model [29,30]. This study determined stability through focus group discussions and in-

terviews with knowledgeable farming households. Inputs included investment items that directly affected the productivity, selling price, and the financial performance of the shrimp–mangrove model, such as the quantity of shrimp and crab stocked and the purchase price. At the same time, the output factors included the model's productivity, revenue, and profitability.

*3.3. Data Collection*

This study applied a participatory approach [31–33] through group discussion tools and in-depth interviews with knowledgeable households conducting shrimp–mangrove model farming. A stratified random sampling technique was applied with 415 observations (Table 2) in the Giao An and Giao Thien districts (buffer zone of XTNP) from August to November 2022.

**Table 2.** Tools of sample observation in the study.

| No. | Tools | Description | Observation |
|---|---|---|---|
| 1 | Interviewing knowledgeable farmers | Leaders of districts, Department of Agriculture and Rural Development, Giao Thuy district, Nam Dinh Provincial Forest Protection Department | 26 |
| 2 | Discussion groups | Covering form (number of discussions in Nam Dinh) Riparian form (number of discussions in Nam Dinh) | 4 4 |
| 3 | Farm household interview | Covering form Riparian form | 250 165 |

Source: Author's compilation (2022).

## 4. Results

*4.1. Summary of Shimp Farming Models*

Table 3 describes some basic physical indicators of the two shrimp farming models. The results showed that in the Giao Thien district, the average water surface area of the households was 4.76 ha while the average mangrove land area was 2448 ha; the largest water surface area was 16.32 ha while the lowest was 1.22 ha; the largest mangrove land area was 10.7 ha and the lowest was 0.68 ha. For farmers in Giao An, the average mangrove area of the households was 4896 ha, of which the average water surface area was 2548 ha. The largest water surface area was 11.56 ha and the lowest was 0.544 ha.

Generally, the ratio of mangrove area to total mangrove land area accounted for 55–65%. However, this ratio often fluctuated in actual production and tended to decrease after aquaculture due to the erosion of the banks and mangrove to the aquaculture ponds (Table 4).

**Table 3.** The structure of the land area of the shrimp–mangrove model.

| Area (ha) | Giao Thien | | Giao An | |
| | Water Surface Area | Mangrove Area | Water Surface Area | Mangrove Area |
|---|---|---|---|---|
| Average | 4.76 | 2.448 | 2.548 | 4.896 |
| Largest | 16.32 | 10.7 | 11.56 | 23.12 |
| Smallest | 1.22 | 0.68 | 0.544 | 1.36 |

Source: Data processing results (2022).

**Table 4.** The ratio of water surface area to total area of mangrove land contracted to households.

| Area | Covering Shrimp–Mangrove Model (*n* = 250) | Riparian Shrimp–Mangrove Model (*n* = 165) | Error |
|---|---|---|---|
| Water surface area (%) | 71.32 | 82.67 | ** |
| Total area (ha) | 6.21 | 5.98 | ns |

Note: **: 99% significance level, ns: not statistically significant by t-test. Source: Data processing results (2022).

The average water surface area ratio ranged from 71.32% to 82.67% of the total mangrove land area of 5.98–6.21 ha/household contracted by farmers (Table 5). The proportion of water surface area in the riparian shrimp–mangrove model group was larger than that in the covering shrimp–mangrove model group. Farmers believed each locality had different regulations, but the water surface area should not be more than 65% of the contracted land. At first, when afforestation occurred, the ratio of the water surface area to the mangrove area was 50:50 on average, but the ponds and ditches eroded over the time of cultivation, so the water surface area became larger.

**Table 5.** Main technical parameters of the shrimp–mangrove model in the study site.

| No. | Shrimp–Mangrove Model Parameters | Giao Thien | Giao An |
|---|---|---|---|
| 1 | Mangrove tree group | Apiculata, Leafy, Avicennia | Apiculata, Corniculatum, Bruguiera |
| 2 | Average tree age (years) | 11.3 [a] | 14.6 [b] |
| 3 | Stocking of shrimp breeds (postlarvae 15) (batch) | 5.3 | 4.8 |
| 4 | Average shrimp density/year (shrimp/m$^2$) | 4.7 | 4.1 |
| 5 | 3rd stocking of crab (pepper crab) (batch) | 4.6 | 4.1 |
| 6 | Average crab density/year (crab/m$^2$) | 2.9 | 2.6 |
| 7 | Shrimp pond improvement (dredging) (year/phase) | 2.7 | 2.3 |
| 8 | Number of harvests (shrimp, crab) (batch) | 12.6 | 12.9 |
| 9 | Average shrimp yield/year (ton/ha) | 85.3 [a] | 92.1 [b] |
| 10 | Average yield of crab/year (ton/ha) | 91.1 | 84.2 |

Note: [a] and [b] are statistically significant by *t*-test at 95% significance. Source: Data processing results (2022).

Regarding the technical parameters, there were many similarities between the riparian and covering shrimp–mangrove models, especially the norms of investment in breeds and farming techniques. Considering the locality, the shrimp–mangrove models in the Giao Thien and Giao An districts had some specific characteristics. In Giao An, the primary mangrove trees were mangroves, tiger trees, and parrots, while Giao Thien had more leafy forests. The farming technique factors, such as the number of shrimp and crab stockings, stocking density of shrimp and crab, etc., were also similar. Plant age and average shrimp yield were higher in Giao An than in Giao Thien (Table 5). However, in both localities, through the results of group discussions, farmers reflected a decrease in shrimp productivity in the last 3–5 years. Previously, the average shrimp yield was 145–185 kg/ha/year, but it had decreased to about 90–125 kg/ha/year.

*4.2. Financial Efficiency of the Shrimp–Mangrove Model*

4.2.1. Costs of Shrimp–Mangrove Model

The survey results showed that the shrimp–mangrove model had two types of costs: fixed and variable costs.

Fixed costs: This referred to a one-time investment cost when designing the shrimp–mangrove model, calculated by the average depreciation of the years of use and operation. In this cost, the main items were investments in the water supply and drainage system for operation, management, and regulation of the volume of water in the pond and harvesting equipment.

Variable costs: These were items for the maintenance and development of the shrimp–mangrove model, such as pond renovation/improvement, investment in plants and breeds, labor costs, food costs, water treatment costs in shrimp ponds, and some other costs.

Pond renovation: The costs of dredging the system of canals and ditches in the shrimp ponds to improve the pond environment, such as treating organic waste at the bottom of the pond, washing alum, and releasing toxic gases and pathogens accumulated in the shrimp ponds from the previous cultivation.

Investment in plants and breeds: These costs included shrimp, crab, fish breeds, and breedings. Although shrimp was the main object, farmers always combined crab and fish in the form of compensatory collection to diversify income from the model.

Labor: For the shrimp–mangrove model, labor costs were mainly invested in the care, management, and monitoring of shrimp growth, especially seasonal labor for water management and harvesting. For the most part, farmers use family labor, while hired labor was used almost exclusively in breed transportation and water quality testing.

Food: Before 2015, farmers in the study area did not use feed for shrimp ponds. However, during this study's survey, the shrimp–mangrove model farmers used foods such as snails, mussels, and mollusks for shrimp and crab. These food sources were exploited by farmers in the wild or purchased locally. However, these food sources were not provided regularly but periodically according to the tide or used as bait for harvesting. According to farmer feedback, the shrimp–mangrove model only added the above food sources when the percentage of shrimp and crab stocked with a high survival rate was high; on the contrary, the shrimp's food source was mainly from the natural environment in the shrimp farming square.

Pond water treatment: Every year, before the new stocking crop, farmers treated the water in the pond, such as killing trash fish with the roots of fish medicinal plants (the main ingredient contained saponins) that were only toxic to fish and not toxic to crustaceans (shrimp, crab).

Other costs: These included harvesting, tools, and equipment for containing, preserving, and transporting shrimp and crab.

4.2.2. Revenue Sources of Shrimp–Mangrove Model

The revenue of the shrimp–mangrove model included shrimp, crab, fish, and mangrove products. This model gave revenue not concentrated at one time but based on batches (collecting shrimp, crab, and fish). After releasing shrimp and crab for the first time, farmers began to harvest shrimp and crab along the water about three months later. At the same time, farmers also stocked up on new shrimp and crab varieties. For the mangrove canopy, the primary sources of revenue were support for mangrove maintenance from the state and harvested mangrove trees.

In previous studies, Walcker et al. (2018) and Tran et al. (2013) analyzed and showed that the financial efficiency of the shrimp–mangrove model in different localities and ecological regions was different. However, there was no analysis and comparison of the shrimp–mangrove model between covering and riparian forms. Therefore, this study focused on the financial performance when comparing the two types of mangrove layouts in the shrimp–mangrove model [34,35].

In 2019–2021, the severe effects of climate change, especially the prolonged heat, drought, and disease of shrimp, affected the efficiency of model farming. The farming conditions

and financial efficiency of the two models were also different. The results in Table 6 showed that most of the cost items did not differ between the two models of shrimp–mangrove farming. Labor costs accounted for 37.4% to 39.8% of total production costs. This was appropriate because the shrimp–mangrove model is practiced extensively, so the techniques are mainly non-mechanized, thus leading to high labor costs. On average, each year, farmers invested about 18.4–20.6 million VND/ha of water surface area for shrimp farming; this investment was suitable for the resources of extensive farming households. The two models' total revenue varied from 51.5 to 54.5 million VND/ha/year, with no statistically significant difference. The proportion of shrimp and crab revenue was almost equal, accounting for 41–51% of the total revenue of the model. Notably, the covering shrimp–mangrove model had a revenue of 24.9 million VND/ha/year, which was higher than the riparian model at only 21.3 million VND/ha/year. Although the total revenue was not different, the profit between the two models differed. The covering shrimp–mangrove model had a profit of about 33.81 million VND/ha/year, which was higher than the riparian model at only about 33.08 million VND/ha/year.

**Table 6.** Financial efficiency of covering and riparian shrimp–mangrove models. (Unit: 1000 VND/ha/year).

| Item | Covering Shrimp–Mangrove Model (*n* = 250) | Riparian Shrimp–Mangrove Model (*n* = 165) | Statistical Error (Significance) |
|---|---|---|---|
| Drains, sewage, nets | 400.78 | 363.75 | ns |
| Rehabilitation of ponds and ditches | 1597.72 | 1752.20 | * |
| Shrimp breeds | 2318.61 | 2437.53 | ns |
| Crab breeds | 2346.56 | 1736.02 | ns |
| Fish breeds | 199.59 | 203.64 | ns |
| Complementary food | 1861.44 | 1456.37 | ns |
| Treatment of ponds and ditches | 532.70 | 430.69 | ns |
| Labor | 7741.94 | 7365.81 | ns |
| Other costs | 3676.90 | 2737.78 | ns |
| Total costs | 20,676.37 | 18,484.16 | * |
| Revenue from shrimp | 24,909.26 | 21,321.61 | * |
| Revenue from crab | 25,380.77 | 26,199.99 | ns |
| Revenue from fish | 4016.25 | 3881.76 | ns |
| Revenue from mangroves [a] | 182.92 | 160.73 | ns |
| Total revenue | 54,489.32 | 51,564.21 | * |
| Profit | 33,812.96 | 33,080.05 | * |

Note: *: 95% significance level, ns: not statistically significant by t-test; [a]: In this study, revenue from forest timber sales was not included because the wood had not yet been harvested; the primary sources of revenue from the forest were pruned wood, support costs of forest land, honey, and other forest products. Source: Data processing results (2022).

In terms of capital efficiency, with 1 VND of farmer investment in the riparian shrimp–mangrove model, farmers would earn 1.79 VND, which was higher profit than the covering model at only 1.64 VND. This result was higher than that reported by Populus et al. (2004), who showed a capital efficiency of 1.21 times [36]. For labor efficiency, in the riparian shrimp–mangrove model, this coefficient was 1 VND investment in labor costs and the farmer earned 4.5 VND in profit; the coefficient for the covering model was 4.367. In terms of profit to total income, in the riparian shrimp–mangrove model, farmers received 1 VND of revenue for which there was 0.6415 VND of profit, while in the covering model, farmers only received 0.6205 VND. In general, the riparian shrimp–mangrove model had many advantages over the covering shrimp–mangrove model, in which the low investment cost was a competitive advantage for farmers.

### 4.3. Factors Affecting the Shrimp–Mangrove Model

The multivariate regression model was used to determine the factors that affected the financial performance of the shrimp–mangrove model. The dependent variable was the profit from the model while the explanatory variables were drawn from the literature review combined with expert and farmer consultation through knowledgeable interviewers and group discussions. There were eight variables: training participation, shrimp breed price, production linkage, the quantity of shrimp breeds stocked, number of stocking times of shrimp breeds, the quantity of crab breeds stocked, price of crab breeds, and age of the person directly managing the shrimp–mangrove model. In the multivariable regression model, the VIF coefficient was also used to test the multicollinearity of the explanatory variables.

The results presented in Table 7a–c summarized the model, showing that the multiple correlation coefficient R = 0.837 (non-zero). This showed that some of the eight tested variables correlated with the model's profitability. The coefficient of determination $R^2 = 0.701$, or about 70% of the variation in profitability, was explained by the variables included in the model, especially the purchasing contract, number of shrimp breeds, number of stockings, number of stocked crab breeds, price of crab breeds, and farming experience of farmers directly managing the model. The remaining 30% of the returned variation was explained by other factors not included in the study in this model.

**Table 7.** (**a–c**) Multivariable regression model parameters.

| (a) | | | | | | |
|---|---|---|---|---|---|---|
| **Model** | **R** | **R Squared** | **Adjusted R Squared** | **Std. Error of the Estimate** | **Dubin–Watson** | |
| 1 | 0.837 [a] | 0.701 | 0.693 | 0.35743 | 1.811 | |

| (b) | | | | | |
|---|---|---|---|---|---|
| **Model** | | **Sum of Squares** | **Mean Square** | **F** | **Sig.** |
| 1 | Regression | 102.268 | 17.045 | 132.367 | 0.00 [b] |
| | Residual | 43.835 | 0.128 | | |
| | Total | 146.103 | | | |

| (c) | | | | | | |
|---|---|---|---|---|---|---|
| **Model** | **Unstandardized Coefficients** | | **Standardized Coefficients** | **T** | **Sig.** | **Statistics VIF** |
| | **Mean** | **Standard Error** | **Beta** | | | |
| Constant | −491.639 | 240.470 | 0.000 | −2.835 | 0.061 | 1.467 |
| Training (X1) | 148.918 | 64.373 | 0.158 | 3.208 | 0.032 | |
| Price of shrimp breeds (X2) | 5.323 | 2.082 | 0.203 | 3.547 | 0.017 | 1.259 |
| Production affiliate (X3) | 790480.427 | 259.996 | 0.239 | 4.216 | 0.004 | 1.513 |
| Amount of shrimp stocked (X4) | 0.003 | 0.000 | 0.503 | 8.789 | 0.000 | 1.689 |
| Number of shrimp stockings (X5) | −70.790 | 41.506 | −0.128 | −2.366 | 0.128 | 1.782 |
| Amount of crab breeds (X6) | 0.018 | 0.004 | 0.379 | 5.784 | 0.000 | 1.521 |
| Price of crab breeds (X7) | 0.161 | 0.018 | 0.847 | 12.028 | 0.000 | 1.634 |
| Experience (X8) | 9.771 | 3.119 | 0.212 | 4.344 | 0.003 | 1.235 |

Source: Data processing results (2022). [a]. Dependent Variable: Y; [b] Predictor: (Constant).

ANOVA Sig analysis results. = 0.000 < 0.05 (error 5%). Therefore, it can be concluded that the given model fit the actual data. In other words, the explanatory variables Xn had a linear correlation with the dependent variable Y with 95% confidence. Of the eight variables included in the test model, three variables were not statistically significant, namely participation in technical training courses, the price or size of shrimp breeds at the beginning of the season, and the number of shrimp stocking times in the crop. However, if

considering the tolerance level of 10%, the number of shrimp stocking times had an inverse effect on the profit (yield) from the shrimp farming model. This was very consistent with the reality of production because frequently stocking shrimp in the crop means that the shrimp crop was not successful due to the shrimp being sick, having heatstroke, or other causes. On the contrary, if shrimp stocking was successful, the farmer harvested without stocking many times.

### 4.4. Financial Stability of the Shrimp–Mangrove Model

The interview results with knowledgeable farmers and the focus group discussions indicated that four input factors affected the stability of the shrimp–mangrove model, including initial stocking quantity, shrimp breed price (size, quality), quantity of crab stocking, and crab breed price (size). The amount of fish and other aquatic animals was not recommended because the fish source was natural, so the cost weight of the breed investment was not significant compared to that of shrimp and crab. Meanwhile, the output factors were shrimp yield, crab yield, shrimp income, crab income, fish income, and total profit of the model.

The data presented in Table 8 show the coefficient of variation (CV) values of the shrimp and crab stocks of the surveyed households in the study area, indicating that the averages were relatively high at 46–53% (shrimp) and 62–76% (crab). This showed that the difference in breed investment for the mangrove–shrimp model among farmers was quite significant. In actual production, due to different conditions and resources, especially financial resources, qualifications, and farming experience, farmers decide to release (invest) a certain number of breeds in a particular area. Households that bought large-sized, high-quality, and high-priced shrimp or crab breeds often stocked at low density, whereas households that bought small-sized, poor-quality shrimp breeds (PCR test) stocked at high density. Farmers reported that stocking at high density compensated for the losses due to shrimp mortality. If shrimp stocking was successful, farmers did not or rarely released crabs.

**Table 8.** Stability of the input factors (%).

| Model | Amount of Shrimp Stocked | Price of Shrimp Breeds | Amount of Crab Stocked | Price of Crab Breeds |
|---|---|---|---|---|
| Covering model | 53.97 | 50.90 | 76.49 | 175.98 |
| Riparian model | 46.38 | 40.91 | 62.61 | 201.04 |
| Average | 50.79 | 46.97 | 72.19 | 185.05 |

Source: Data processing results (2022).

In contrast, some other farmers, when they failed to release shrimp, released crabs to compensate for the model's revenue. There was also the concept of releasing crabs as a source of regular income (crab fishing) in order to help farmers cover their living expenses, while shrimp stocking provided concentrated income in batches. This was the main reason for the households' significant variation in shrimp breed stocking.

Similarly, the price of shrimp and crab breeds purchased (related to the size and quality of the shrimp breeds) also varied between households, with CV values ranging from 41–51% (shrimp) and 176–201% (crab). In particular, the purchase price of crab was very different between households. The main reason was that farmers combined concentrated investments and took advantage of local resources, thus investing evenly. During the beginning of the production season, households released large quantities of small crabs.

During production, households could catch or buy wild large-sized crabs from crab catchers. These crabs were still added to the farming square. These crabs had a fast harvest time, high yield, and little loss. However, the quantity was limited and unstable depending on natural and seasonal crab catchers, so it was impossible to invest simultaneously in this resource. This was the reason for the significant fluctuation in crab investment among households implementing the shrimp–mangrove model.

In general, when assessing the stability of a production system such as the shrimp–mangrove model, it is necessary to consider two aspects: (1) variation among producing households at the same time of cultivation; (2) variation within the same household or group of households in a time series of cultivation. This study's analysis was limited to the variation among producing households in the same period. The above results showed the coefficient of variation of input factors (shrimp, crab breeds) of the shrimp–mangrove model. This proved that the stability of the model input was not high, which could affect the output stability and the model as a whole (Table 9).

**Table 9.** Stability of the output factor (%).

| Model | Shrimp Yield | Crab Yield | Shrimp Revenue | Crab Yield | Fish Yield | Total Profit |
|---|---|---|---|---|---|---|
| Covering model | 78.518 | 73.854 | 43.626 | 71.83 | 43.296 | 57.497 |
| Riparian model | 65.34 | 60.621 | 29.216 | 21.626 | 41.591 | 25.344 |
| Average | 74.987 | 68.376 | 40.293 | 56.496 | 42.482 | 47.465 |

Source: Data processing results (2022).

In addition to the input factors, the output factors such as shrimp yield, crab yield, shrimp income, crab income, and profit of the whole model were also analyzed for fluctuations in the stability assessment of the model. The results of the data analysis presented in Table 9 indicated that shrimp productivity, crab productivity, and income from crabs fluctuated relatively highly among producing households (CV values were 75%, 68%, and 56%, respectively). Meanwhile, income from shrimp and fish was less volatile (CV values of 40% and 42%). The model's final total return had a coefficient of variation of 47%.

The income or profit of the shrimp–mangrove model is a composite interaction between the selling price and shrimp yield. In principle, the selling price of shrimp is uniform and fair among households, only differing in the type and quality of harvested shrimp. In addition, the selling price of shrimp also depends on the size of the shrimp (large-sized shrimp have a higher selling price than small-sized shrimp).

Thus, shrimp productivity was the decisive factor in whether the shrimp–mangrove model was stable. Productivity is a composite indicator resulting from the interaction between input factors such as breed, labor, and rearing environment (soil, water). At the same time, this interaction is complex, challenging to measure, and heavily influenced by other exogenous factors. In this study, the productivity of shrimp and crab (two primary sources of income) had a high coefficient of variation, so it can be confirmed that the stability of the model was not high (Table 9).

*4.5. Advantages and Disadvantages of the Shrimp–Mangrove Model*

Each locality had its advantages and disadvantages for the shrimp–mangrove farming model. In general, the main advantages were the nature of the model and the comparative advantage of natural conditions, such as soil, water, low investment, use of agricultural labor, diversified income sources, less environmental pollution, and limited risk. At the same time, the main difficulties of the model were low productivity, limited infrastructure system, farming techniques, breed quality, disease, and impacts of climate change and environmental change (Table 10).

Based on an analysis of the advantages and disadvantages, some solutions are proposed to improve the model's efficiency, such as designing and renovating ponds, certifying safe products, providing technical support, improving breed quality, improving farmers' capacity, and connecting farming households and collectors or traders to achieve sufficient output for the market.

**Table 10.** Advantages and disadvantages of the shrimp–mangrove model in the Giao Thuy district.

| SWOT | Opportunities (O) O1: Diverse natural food sources; O2: Combined culture with many other aquatic species; O3: Applying ecological and organic certification standards; O4: The price is stable and tends to increase; O5: In line with the trend of sustainable development. | Threats (T) T1: Effects of climate change and environmental pollution; T2: Breed quality is not stable; T3: Risk of disease; T4: The policy on mangrove land allocation has changed (not yet facilitated to take advantage of existing resources). |
|---|---|---|
| Strengths (S) S1: Large farming square area; S2: Multiple revenue sources in the model (product diversity); S3: Low investment costs compared to intensive farming model (no cost of water change); S4: Take advantage of family labor; S5: Safe, environmentally friendly products | Combination of O+S O1,2+S1,2,3: Improve the farming square to increase natural food sources and, at the same time, raise many other species; O3,4,5+S4,5: Apply standards of organic shrimp certification to promote high-quality shrimp products. | Combination of S+T S1+T1: Square farming design ensures technical requirements to limit the impact of climate change; S3,6+T2,3: Select good quality shrimp breeds to increase survival rate and reduce disease; S4,5+T4: Have a clear orientation and plan for land use (contracting). |
| Weaknesses (W) W1: Shrimp productivity is low compared to intensive farming; W2: The square design is not suitable (low water level in the pond); W3: Sludge settles quickly on the pond bottom, costly to improve; W4: Farming techniques are still limited; W5: Underdeveloped transport infrastructure. | Combination of O+W O1,2+W1,2,3: Square design according to technical requirements to take advantage of natural food sources and combined farming; O3,4,5+W4,5: Technical support for farming households, applying ecological shrimp certification standards to improve model production efficiency. | Combination of T+W W1,2,3+T1: Design the farming square according to technical requirements to reduce the risk of impacts of climate change; W4+T2,3: Capacity building for farmers in square management of disease n shrimp; W5+T4: Links between farming households to obtain large output and increase the selling price. |

Source: Author's compilation (2022).

## 5. Discussion

This study shows the financial efficiency of shrimp farming using covering and riparian mangrove models. In both models, variable costs accounted for a large proportion of the production cost structure, in which the covering model had higher variable costs due to differences in labor, feed, and pond treatment costs. In the total cost structure, labor costs accounted for the highest proportion in both farming models (about 33% in the covering model and 38% in the riparian model). This result was consistent with the studies by Bosma (2014), Johnston (2000), and Tran (2013). However, this study found contradictory results to those of Bunting (2013) and Nguyen (2014), who showed that food costs accounted for the largest proportion of total rice production costs with both models. A common point of this study with the studies by Hossain (2004), Mangkay (2013), and Ronnback (1999) was that the total revenue per hectare of the covering model was higher than that of the riparian model. The reason was that the income from shrimp and mangroves in the covering model was larger than that in the riparian model, and the cost of pond renovation in the covering model was smaller. This finding was consistent with the observations of Bunting (2013), Van Oudenhoven (2014), and Quyen (2021).

In general, the shrimp-mangrove model is affected and governed by many factors, including breeds, farming techniques, market prices, and environmental factors such as soil, water, and salinity. However, in this study, five factors had a clear impact and influence on the financial performance (profit) of the model, including production linkages, number of stocked shrimp breeds, number and size of stocked crabs, number of stockings, and farming experience of the person directly managing the shrimp–mangrove model. Identifying these factors is the basis for proposing solutions to improve the financial efficiency of the model. The following variables were specifically correlated with profit from shrimp in the model:

Production contract: This variable had a positive Beta coefficient, indicating that it positively impacts the shrimp–mangrove model's profitability. In other words, when farmers implement the shrimp–mangrove model with production connections (forms of association include an agreement to provide breeds and input materials or to purchase and consume output with enterprises or traders based on a predetermined purchase price) will result in increased profit compared to unaffiliated farmers. This is consistent with reality because an affiliation with input suppliers (breeders) will support farmers through pricing policies, after sales, and quality assurance (familiar and reputable customers). Thereby, expenses and production costs are reduced, contributing to improved profits. On the contrary, if farmers do not cooperate, they will have to buy breeds of unknown origin; sometimes, the product is not of guaranteed quality. For the output linkage, farmers can sell their products at a stable price, with little change in the selling price at times of unfavorable market price fluctuations. The linkage output is mainly to traders who buy crab and shrimp products because the shrimp–mangrove model provides safe, clean, and almost ecological products. The crab and shrimp obtained in the shrimp–mangrove model have better basic (meat) quality than the shrimp and crab obtained from intensive farming;

Amount of shrimp stocked: Similarly, stocking of shrimp variable positively impacts the shrimp–mangrove model's profitability. When households stock high-density shrimp breeds, they will have a better chance of earning profits than low-density stockings. In production, in general and in the shrimp–mangrove model in particular, breeds/seeds play a critical role in deciding the model's success, failure, productivity, and profitability. When the breed and seed source is good, proper farming techniques will give high yields, which leads to high profits. In contrast, when the breed's source is not good, disease occurs, the growth and development of shrimp are not achieved, and production costs increase, leading to reduced profits. For the shrimp–mangrove model, shrimp breeds were rarely stocked in the past; households only needed to open square culverts to allow water in. Subsequently, the source of shrimp breeds also followed into the square and the shrimp ate natural food under the mangrove canopy. Plankton in the shrimp square grew and was harvested by farmers.

Since the 1990s, the sources and amounts of natural shrimp breeds are no longer as abundant as before and their vitality has also weakened. Thus, natural shrimp breeds find surviving in the shrimp square environment challenging because of competition with many other aquatic animals. Therefore, additional stocking of shrimp breeds for the shrimp–mangrove model is recommended. When stocking high-density shrimp according to the extensively used threshold (the average survey results were 3–5 shrimp/m$^2$, with a maximum of 6 shrimp/m$^2$), stocking is performed once if the environment is good and the shrimp grow and give high yield. If the environment is unfavorable, the shrimp will die and farmers will have to stock many times. However, the stocking density of shrimp is limited to a specific limit, ensuring that the living environment is not polluted, the natural food source is guaranteed, and shrimp productivity is reached. If the density is too high, the shrimp square will be contaminated and the shrimp will die, thus reducing the harvest. Because in the shrimp–mangrove model the fan system to provide oxygen is unavailable, pond water treatment interventions are not implemented and the habitat remains natural.

Amount of crab breeds stocked: This variable also positively affects the model's profitability. When other conditions are normal (no hill), the amount of crab breeds released into the shrimp–mangrove square (pond) helps to increase the profitability of the model. In the past, when raising shrimp, farmers did not or rarely released crab into the pond because there was mutual eating when one of the two molted. Ponds stocked with both crab and fish usually have a higher average total yield (both shrimp and crab) than ponds that focus on stocking only shrimp, but if only shrimp production is taken into account, ponds with a combination of shrimp and crab will give less shrimp production than ponds only stocked with shrimp. In addition, ponds with a combination of shrimp and crab have a stable yield and are less risky than squares with only shrimp. Recently, due to pollution of the farming environment (water, breeds), farmers realized that the square stocked with

shrimp only would be riskier than the square stocked with shrimp and crab. In the case of shrimp encountering problems, the crab income will compensate. Therefore, farmers added crab to the shrimp–mangrove model square. Through reasonable supplementation with crab food and appropriate water management techniques, farmers avoid situations where crabs and shrimp cannibalize each other when molting.

Price of crab breeds: The price of juvenile crab variable is positively correlated with the profitability of the shrimp–mangrove model, which shows that when other factors are kept constant, the higher the price of crab breeds, the higher the profit of the shrimp–mangrove model. In essence, this is explained by the price of crab breeds varying with size; a high crab breed price means that large breeds are stocked in the square. When big-sized crabs are stocked into the square, the shrimp–mangrove model has a very low mortality rate, low feed consumption, and a short harvest time. For a long time, shrimp–mangrove model farmers were aware of and understood this rule. However, the amount of crabs in the wild is insufficient and their sizes are very different, so it is difficult to achieve large-scale stocking with uniformly large-sized crabs bought from crab catchers for release into the model's shrimp square. The shrimp–mangrove model is only suitable for small-scale production with many crops in the year.

Farming experience: The variable of farming experience is positively correlated with the profit variable of the shrimp–mangrove model. This indicates that if the variables (other factors) are fixed, long-term experience in shrimp–mangrove farming will likely increase the model's profits. It is possible that through the process of farming, farmers accumulate experience, understand environmental conditions and the nature of shrimp ponds, and gradually improve farming techniques, thus contributing to improving the financial efficiency and profitability of the model.

Regarding the stability of shrimp farming models, previous studies and reports by Ronnback (1999) and Ruitenbeek (1994) suggested that agroforestry models such as the shrimp–mangrove model are low risk, but their stability had not been considered. Meanwhile, the results of analyzing the inputs and outputs of the shrimp–mangrove model in this study showed that the model had low stability. Therefore, although there is little risk, the stability of the shrimp–mangrove model is uncertain and always depends on environmental conditions that vary from place to place [34,35].

The results of this study once again provide information for the assessment and recognition of the true nature of the shrimp–mangrove model. If the variation (inputs such as breed stocking, outputs such as productivity, revenue) among model farming households is considered an indicator of the model's stability, the results of this study confirmed that the current shrimp–mangrove model is not highly stable. However, considering the level of risk, this model is less risky because the investment capital for production is not high (compared to the intensive farming model and high-density shrimp farming), while the model's revenue sources are relatively diverse. Farmers can collect crab or fish to compensate if households lose shrimp revenue [36]. The widely distributed income helps farmers to invest in the crop, which is suitable for those with little production capital. From a social and environmental perspective, the shrimp–mangrove model takes advantage of family labor (profit), creates a healthy environment, and helps in greening the mangrove area. This finding is in line with studies by Populus (2004), Tran (2012), and Nguyen (2014).

## 6. Conclusions

The results of this study indicated that the current shrimp–mangrove model has low stability and decreased financial performance indicators, such as shrimp productivity, revenue, and profit, compared to previous years. In the two forms of shrimp–mangrove arrangement, riparian and covering, the main income sources are shrimp, crab, and fish. The target shrimp revenue and overall profit of the covering shrimp–mangrove model were higher than those of the riparian shrimp–mangrove model. Meanwhile, other indicators, such as total cost, revenue from crabs, revenue from fish, and total revenue, showed no difference between the two models.

There are many factors affecting the financial performance of the shrimp–mangrove model. Among them, the purchase contract, number of shrimp breeds, number of stocking times, number of stocking crabs, price of crab breeds, and farming experience are significant contributors. In terms of the stability of financial expenditure, the shrimp–mangrove model currently has large fluctuations in inputs, such as the amount of shrimp and crab stocked, breed prices, and outputs, such as average productivity and profit among farm households under the same cultivation conditions.

Due to the diversity and complexity of the model farming process, households have different resources (qualifications, farming techniques, investment capital, labor), so the fluctuations of inputs and outputs among households are quite large. This affects the stability of the model. Therefore, although the shrimp–mangrove model is extensively practiced, it still needs a uniform production process suitable to the actual conditions and local resources in order to ensure the effectiveness and stability of the model. This is also necessary to encourage the development of this model now and in the future.

Therefore, in order to ensure the effectiveness and stability of the model, solutions such as renovating the ponds, certifying safe products, providing technical support, improving the quality of breeding stocks, improving human capacity, and building linkages (buying/selling) between farming households and local businesses or traders are essential. The study results provide a valuable reference for the agriculture and fisheries sector and local leaders in the development and planning of fishery production in particular and coastal land use in general.

**Author Contributions:** Conceptualization, N.T.M.; Methodology, T.T.L.H.; Validation, T.T.D.; Writing—review & editing, D.D.T. All authors have read and agreed to the published version of the manuscript.

**Funding:** This research is funded by the National Economics University, Hanoi, Vietnam.

**Institutional Review Board Statement:** Not applicable.

**Informed Consent Statement:** Not applicable.

**Data Availability Statement:** Not applicable.

**Conflicts of Interest:** The authors declare that the research was conducted in the absence of any commercial or financial relationships that could be construed as a potential conflict of interest.

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
