# Peer review of "An Empirical Study of Financial Efficiency and Stability of Shrimp–Mangrove Farming Model in Nam Dinh Province, Red River Delta, Vietnam"

_sustainability, doi:10.3390/su15076062_

Round 1
Reviewer 1 Report
Dear authors,
After reading the submitted first version of your manuscript I’m listing some comments, questions, doubts, and suggestions that I believe are worth to be mentioned.
The aim of your work is to evaluate the effectiveness and stability of the shrimp-mangrove farming model in the Nam Dinh Province, Red River Delta, Vietnam. The methodology includes different techniques such as a cost-benefit analysis model (to evaluate the financial performance and stability of the shrimp farming model) and a SWOT analysis (to identify the model’s opportunities and threats).
· In terms of the research design, the work is empirically average-based. It is suggested that:
· Your results and findings need to be “polished” to be conveyed better to the readers.
· A better theoretical framework of the multiple regression analysis should be formulated. Did you consider introducing some non-linearity or a climate change proxy into the model? You already provided evidence that there is a strong connection between climate change and shrimp production. Which of the two assessed models resulted more efficient: the riparian form or the covering form? Also, you need to mention the post-estimation test results for the validation of your model.
· In terms of form and presentation:
· Subsection 4.1 should be better placed at Section 2 since is the historical description of the model in the study site.
· Policy implications, deriving from the work’s findings, in terms of sustainability are missing.
· Language style needs to be revised! Typing errors need to be checked! (line 320, the name of the multicollinearity test is wrongly typed for example!)
My best regards!

Author Response
Dear Reviewer
The author is very grateful to the Reviewer for appropriate and constructive suggestions and for their proposed corrections to improve the paper. We have addressed all the issues raised and have modified the paper accordingly. Below is a summary of the changes we performed and our responses to the reviewers’ comments and recommendations. The modifies have been placed and marked in red color text in the revision of manuscript.
Please see the attachment with responses of authors.

Reviewer 2 Report
The abstract is well structured and critically defines the rationale, methodology, findings, and conclusion. The study is valuable for the agriculture as well as fisheries sector particularly for coastal land use. However, the study had failed to show any significant changes in terms of the monetary gain due to low productivity and fluctuation in market prises of different inputs. Some quantifiable unit is required to incorporate in this section as highlighted in the manuscript.
KEYWORDS
Rightly reflected the study main key points but the words used in title may be excluded.
INTRODUCTION
The study adequately elaborated and highlighted the need of the region. However, some necessary incorporations are suggested as follows:
MATERIALS AND METHODS
Experimental details are described in detail and presented nicely. However, some more clarification particularly statistical analysis is to be incorporated. More clarity is to be on costs incurred in production. One Table must be included on different unit wise cost of the production.
RESULTS
The result is nicely synthesized using findings in figures with statistical point of view of the treatments.
DISCUSSION
The discussion is not described in this Manuscript. It should be incorporated as per the format of the journal..
CONCLUSION
The conclusion is sufficiently brief and highlights the possible outcome.
REFERENCES, TABLES AND FIGURES: Kindly follow the Journal style.

Author Response
Dear Reviewer,
The author is very grateful to the Reviewer for appropriate and constructive suggestions and for their proposed corrections to improve the paper. We have addressed all the issues raised and have modified the paper accordingly. Below is a summary of the changes we performed and our responses to the reviewers’ comments and recommendations. The modifies have been placed and marked in red color text in the revision of manuscript.
Please find attachment with responses of authors to the comments.

Reviewer 3 Report
This is a strong, well-written and argued paper that could be published in its current form. I strongly support it. My only comment is that "sustainability" should include being able to attract younger shrimp farmers. I am not sure that many of the smaller or even average plots will bring in enough income to do that. Some reference to this problem - which suggests that future farm size and capital intensity will grow - is perhaps warrented.
Author Response

(The authors gave the same response as above.)

Round 2
Reviewer 2 Report
Thank you very much for your response, found out improvement in the revised version of the Mns.